# Age and Sex-Related Associations between Marital Status, Physical Activity and TV Time

**DOI:** 10.3390/ijerph19010502

**Published:** 2022-01-03

**Authors:** Timothy Gustavo Cavazzotto, Natã Gomes de Lima Stavinski, Marcos Roberto Queiroga, Michael Pereira da Silva, Edilson Serpeloni Cyrino, Helio Serassuelo Junior, Edgar Ramos Vieira

**Affiliations:** 1Department of Physical Education, Midwestern Parana State University, Guarapuava 85040-167, PR, Brazil; consultoriacvzt@gmail.com (T.G.C.); queirogamr@hotmail.com (M.R.Q.); 2Graduate Program in Health Sciences, Center of Health Sciences, State University of Londrina, Londrina 86039-440, PR, Brazil; 3Graduate Program in Public Health, Faculty of Medicine, Federal University of Rio Grande, Rio Grande 96203-900, RIG, Brazil; mpsilva@furg.br; 4Department of Physical Education, Faculty of Physical Education and Sport, State University of Londrina, Londrina 86057-970, PR, Brazil; edilsoncyrino@gmail.com; 5Department of Sports Science at the State University of Londrina, Londrina 86057-970, PR, Brazil; heliojr@uel.br; 6Department of Physical Therapy, Florida International University, Miami, FL 33179, USA; evieira@fiu.edu

**Keywords:** exercise, marriage, sedentary behavior, gender, age groups, behavioral risk factor surveillance system, health promotion

## Abstract

Marital status mediates an association between physical activity (PA) and TV time with health outcomes. However, population-based studies have revealed that the health effect of marriage or divorce is age-dependent and differs between women and men. The study aimed to identify the age and sex-related associations between marital status with PA and TV time. We used data from Vigitel, an annual telephone survey started in 2006 in Brazil. We applied a complex sample logistic regression model to estimate the odds for PA and TV time comparing marital statuses according to age and sex subgroups, independent of obesity, hypertension, diabetes, self-assessed poor health, and smoking. Our sample included 561,837 individuals from 18 to 99 years, with a TV time > 3 h/day (prevalence = 25.2%) and PA > 150 min/week (prevalence = 35%). Later, we divided our sample in seven age groups by marital status and sex. Compared to single individuals, married men and women were less likely to watch TV more than 3 h/day in participants >30 years old. When compared to single, married participants were less likely to do more than 150 min of PA/week at younger age groups. Married women older than 40 years were more likely to do more than 150 min of PA/week than the single ones, while there were no differences among married men by age group. In conclusion, our study suggests that the investments in public policies to encourage the practice of PA and reduction of TV time could be based on the marital status, sex, and age, prioritizing less active groups.

## 1. Introduction

Regular physical activity (PA) practice is associated with good health outcomes, preventing the development of many of the most common non-communicable diseases [1]. On the other hand, sedentary behavior, which is sitting or reclining at an energy expenditure of 1 to 1.5 basal metabolic rates, has been associated with all-cause and cardiovascular mortality, independent of PA levels and body mass index (BMI), independently of moderate to vigorous PA [2]. Therefore, the World Health Organization (WHO) provided guidelines on PA and sedentary behavior, recommending to adults to meet between 150–300 min of moderate-intensity PA weekly. While there is not enough evidence on the quantitative threshold of sedentary behavior, it is recommended to limit the amount of time as much as possible [3]. PA and sedentary behavior (SB) are independent behaviors influenced by several factors [4], which might not be the same for both PA and SB. A critical correlate is the marital status, which has been associated with health benefits and longevity [5].

Recent studies investigating health influence of marital status revealed that married adults have lower morbidity, mortality risks, mental disorders, and suicide risk than unmarried people [6,7,8]. According to a US population-based study, marriage was associated with reducing substance use, such as tobacco, alcohol, and cannabis, compared to single and divorced/separated men [9]. In another study, Kim et al. found that living without a partner results in lower odds of smoking and drinking, while blood pressure and glycohemoglobin are lower in married people [7]. These observations might be explained by the theory of social selection and the theory of social causation [10], which states that healthier people are more likely to get married and stay married longer. In contrast, married individuals showed lower fitness levels than single individuals [11], but there are differences between men and women. A recent population-based survey investigating health-related factors in 59,402 Brazilians found that adults with a sexual partner have 1.2 higher chances of obesity [12]. Therefore, there is some contradictory findings on the influence of marital status on physical activity and sedentary lifestyle, and particularly differences between age and sex subgroups.

More than a fourth of the world population is inactive [13] and spend substantial time in sedentary behaviors [14,15]. Marital status [11], age [16], and sex [17] are associated with health status. A prospective study investigated the association between marital status transitions and fitness level changes in men and women [11]. The study found that men transiting from single to married reduce their fitness levels. At the same time, women in this transition did not change their fitness level. However, women who remained single increased their fitness levels, supporting prior findings that indicate that married individuals practice less PA. However, there seem to be differences between men and women as well as among age groups. O’Donoghue et al. [16] conducted a systematic review to identify associations between sedentary behavior and various outcomes in adults aged 18–65 years. The review found that 14 of the 20 studies found a positive relationship between age and sedentary behavior (i.e., the older the person, the more sedentary).

Contradictions on the relationship between marital status and PA were found in a recent review of reviews. The association was deemed inconclusive in five reviews, no association was found in three reviews, and a negative association was found in one review [18]. However, those reviews analyzed specific subgroups (e.g., only adults without gender separation, only older women, etc.), which limits the analytical power to compare differences between sexes and age groups. In addition, if there are differences on PA and SB across different marital statuses, age groups and sex, these outcomes could guide government investments to reduce SB and increase PA in the most inactive groups. Thus, the current study aimed to evaluate the association between marital status, PA, and sedentary behavior (i.e., TV time) by age and sex groups.

## 2. Materials and Methods

### 2.1. Sample

We used data from the Surveillance of Risk and Protection Factors for Chronic Diseases by Telephone Survey—VIGITEL. It is a telephone survey that started in 2006 and it is annually performed in all Brazilian state capitals and the Federal District to identify the frequency, distribution, and progression of the main risk factors for chronic non-communicable diseases [19,20,21]. We used the data from 2009 to 2019 because some variables of our interest were only added in 2009.

The sampling procedures aim to obtain probabilistic samples of the adult population (≥18 years of age) living in households with at least one phone. The Vigitel system established a minimum sample size of approximately 2000 individuals in each city to estimate the frequency of the main risk factors for chronic non-communicable diseases (hypertension, diabetes, and dyslipidemia), utilizing the rake method to perform the weight calculation for the sample expansion [22]. The Vigitel system establishes a confidence coefficient of 95% and maximum error of 2% points, while an error of 3% points is expected to specific estimates, according to gender and assuming similar proportions of men and women in the sample [23]. First step of the Vigitel sampling consists of drawing at least five-thousand telephone lines peer city. Second step is to draw one of the adults residing in the selected household, which is performed after identifying, among the drawn lines, those that are eligible for the survey.

The attributed weight to each individual takes into consideration two factors: (a) the number of phone lines in the household, since this factor aims to correct the greater chances from households with more than one telephone line had to be selected for the sample; and (b) the number of adults at the respondent’s residence. The product of these two factors provides a sample weight that allows us obtaining a reliable estimation from the adult population in each city. A more detailed information about the sampling methods can be found elsewhere [19].

A specialized company carried out the telephone interviews from Vigitel. The team responsible for the survey involved interviewers, monitors, supervisors, and a general coordinator, varying the number of team members according to the year of the survey. In addition, all the participants of the project received prior training and were supervised during the operation of the system by researchers, and by technicians from the Health Surveillance Secretariat of the Ministry of Health (SVS/MS). The Vigitel questionnaire includes short and simple questions on demographics and socioeconomic characteristics, feeding patterns and physical activity, self-reported weight and height, tobacco and alcohol consumption, and self-evaluation of health status.

### 2.2. Physical Activity and TV Time

To estimate PA, we used the following questions: “In the last three months, did you perform any exercise or sport?”; “What is the main type of physical exercise or sport that you practiced?”; “Do you do the exercise at least once a week?”; “How many days a week do you usually practice exercise or sport?”; “On the days you practice physical exercise or sport, how long (minutes/day) do you perform these activities?”; with the responses to these questions, we estimated the prevalence of leisure time physical activity >150 min/week, which is the minimum recommendation from World Health Organization [3].

We used the following questions to evaluate TV time: “How many days a week do you watch TV?”; “How many hours do you watch TV per day?”. There were eight alternatives to answer, ranging from “less than one hour a day” to “more than six hours a day”, including “I don’t watch TV”. With the responses to this question, we estimated TV time.

### 2.3. Covariates

The predictive model considered the following covariates: sex, age, obesity, hypertension, diabetes, self-evaluation of poor health, smoking, and year of data collection. Obesity was evaluated based on the calculation of Body Mass Index (BMI ≥ 30 kg/m^2^ = obesity) [24] using the answers to the following questions: “What is your weight?”, and “What is your height?”.

We included all covariates based on the literature. Correlates and determinants of physical activity and sedentary behavior have been discussed in several previous studies, such as age (inversely), male, overweight (inversely), non-communicable diseases [4,25,26].

Hypertension was evaluated based on the following question: “Have any physician ever told you that you have high blood pressure/hypertension?”. Diabetes was assessed based on the following question: “Have any physicians ever told you that you have diabetes?”.

Self-assessment of health status was evaluated using the following question: “How would you rate your health status?”. Health was classified as poor if they answered “bad” or “very bad”. Smoking was assessed by asking “Do you smoke?” independently of amount, frequently or the duration.

### 2.4. Statistical Analysis

The descriptive results were presented as relative frequency and confidence intervals. Differences in prevalence between sex and age groups were tested by chi-square and confirmed by no overlapping confidence intervals. A complex sample logistic regression model was applied to estimate the odds ratio for PA > 150 min/week and TV time > 3 h/day by marital status stratified by age groups and sex. We used as reference group single and divorced, adjusting by year of data collection, obesity, hypertension, diabetes, health status, and smoking as covariates. All results were estimated using a complex sample model in SPSS software version 25.0 (SPSS Inc., Chicago, IL, USA) applying strata and sample weight provided in the database.

## 3. Results

This study included 561,837 individuals (62% were women). Drawn, eligible, and reported phone lines of Vigitel from 2009 to 2019 are presented at the Appendix A (Table A1). The prevalence of obesity was 18%, diabetes 7%, and hypertension 25%. Smoking was more prevalent in men than in women (14.2% vs. 9%); women had higher prevalence of self-reported poor health than men (5.8% vs. 3.3%) (Table A2). The prevalence of TV time > 3 h/day was 25.2% with no sex differences, while PA was significantly higher in men than women (43% vs. 28.2%) (Table 1).

When stratified by sex and age, the prevalence of TV time > 3 h/day was higher in men compared to women at most age groups: 31–40 years (26.2% vs. 21.5%), 41–50 years (24.8% vs. 21.9%), 61–70 years (32.1% vs. 26.2%), 71–80 years (34.5% vs. 28.2%), and 80+ years (37.1% vs. 27.9%). Single and married men had lower prevalence of TV time > 3 h/day than divorced and widowed ones, while married women had lower prevalence of TV time > 3 h/day than single, divorced, and widowers.

Prevalence of PA > 150 min/week reduced in both men and women from the 18–30 to the 80+ age group (men: 58.6% vs. 22%; women: 32.3% vs. 12.6%, respectively). Men had a higher prevalence of PA > 150 min/week than women in all age groups. When stratified by marital status, single men had a higher prevalence of PA > 150 min/week than married and divorced ones (54.4% vs. 34.3% vs. 35.1%, respectively), while widowers had the lowest prevalence of PA > 150 min/week (26%). Single women had a higher prevalence of PA > 150 min/week than married women (30.9% vs. 28.1%, respectively), while both single and married women had higher prevalence of PA > 150 min/week than divorced (24.7%) and widower (20.4%) women. Men had higher prevalence of PA > 150 min/week than women independently of marital status (Table 2).

Table 3 presents the odds ratio of watching TV more than 3 h/day and of practicing PA more than 150 min/week by age group, marital status, and sex. Married individuals were less likely to watch TV more than 3 h/day compared to singles in the age groups of 31–40 (men: OR = 0.79 vs. women: OR = 0.75), 41–50 (men: OR = 0.66 vs. women: OR = 0.68), 51–60 (men: OR = 0.80 vs. women: OR = 0.66), 61–70 (men: OR = 0.73 vs. women: OR = 0.65), 71–80 (only women: OR = 0.84), and 80+ (only women: OR = 0.59). Also, married individuals had lower odds of watching TV > 3 h/day than divorced ones in the age groups of 18–30 (men: OR = 0.65 vs. women: OR = 0.69), 31–40 (men: OR = 0.73 vs. women: OR = 0.72), 41–50 (only women: OR = 0.71), 51–60 (only women: OR = 0.81), 61–70 (men: OR = 0.76 vs. women: OR = 0.79), and 71–80 (only women: OR = 0.59).

Regarding PA, married individuals had lower chances of practicing PA > 150 min/week compared to singles in the age groups of 18–30 (men: OR = 0.59 vs. women: OR = 0.74), and 31–40 (only men: OR = 0.79). However, chances of meeting PA > 150 min/week were higher in married individuals than in singles in the age groups of 41–50 (only women: OR = 1.29), 51–60 (only women: OR = 1.36), 61–70 (only women: OR = 1.44), and 71–80 (only women: OR = 1.36). Married individuals had higher chances of meeting PA > 150 min/week than divorced ones on the age groups of 18–30 (only women: OR = 1.26), 31–40 (only women: OR = 1.11), 41–50 (men: OR = 1.14 vs. women: OR = 1.21), 51–60 (only men: OR = 1.24), 61–70 (men: OR = 1.24 vs. women: OR = 1.21), and 71–80 (only women: OR = 1.44). Widowers were not included in Table 3 because they are more heterogeneous group, especially at the younger age groups. Table A3 and Table A4 describes additional comparisons of the chances of watching TV > 3 h/day and meeting PA > 150 min/week between different marital statuses and sex using single (Table A3) and divorced (Table A4) as references.

## 4. Discussion

Our study aimed to identify the magnitude of independent associations between marriage and other marital statuses with physical activity and TV time. The main findings of this study were:(a)TV time > 3 h/day was more prevalent in younger men and in older women;(b)The prevalence of PA > 150 min/week decreased with age, and in general, men were more active than women;(c)married men and women had lower odds of watching TV > 3 h/day than single and divorced ones in most age groups, especially women;(d)married men and women in younger age groups were less likely to practice PA for more than 150 min/week levels than single ones, but among those older than 40, married women had higher odds of meeting PA levels than single ones;(e)married women had higher chances of practicing PA > 150 min/week than divorced ones in most age groups, while middle aged (41–70 years) married men had higher chances of practicing PA > 150 min/week than divorced middle-aged men.

More than a fourth of the individuals watched TV more than three hours a day. Three or more hours of TV time/day is associated with increased mortality risk [27]. Current lifestyles in society provide many opportunities for leisure-time sedentary behavior like watching too much TV. Elevated levels of sedentary behavior increase the risk of metabolic dysfunctions such as dyslipidemia and insulin resistance, and have deleterious effects for bone and cardiovascular health [28]. Each 1-h increment in total sitting time increases the risk of having sarcopenia by 33%, however, only TV time was associated with lower levels of lean body mass [29]. Watching TV is not only a sedentary behavior but is also linked to other health risk behaviors such as unhealthy diet with higher consumption of energy-dense foods and fewer fruits and vegetables [30].

Interestingly, TV time > 3 h/day was more prevalent in younger men and in older women. Nathanson et al. [31] provides a theoretical framework on the topic, where men are more likely to watch TV for goal-directed purposes, such as viewing the news while women are more likely to watch TV as a substitute for social interaction. As women tend to live longer than men [32], it is likely that women increase TV use to replace their partner’s social interaction.

Our results showed that 35% of individuals practiced more than 150 min/week of leisure-time PA. In contrast, Rech et al. [33] found that only 18.8% of the individuals in their study practiced more than 150 min/week of leisure-time PA, probably because the study was conducted in a single capital city. Similarly to our findings, women were less active than men in the study conducted by Guthold et al. [13]. They found that lower levels of physical activity were more prevalent in women in Latin America and the Caribbean compared to other continents. This might be due to differences between sexes because of factors such as competition and challenges, or intrinsic factors such as self-efficacy and social values regarding physical appearance [34]. Recent finds suggest that PA levels have a dose-response relationship with reduced risk of all-cause mortality [27,35,36], while sitting time increases the risk of all-cause mortality, PA mitigates the increased risk [27]. Single men and women had higher prevalence of PA > 150 min/week than married counterparts. This was not expected based on our hypothesis that married individuals would have a healthier lifestyle. However, this might be because married individuals have reduced free time to practice leisure-time PA. Further investigations are required to confirm if married individuals indeed have reduced free time and, if so, comprehend its causes and find alternatives for active lifestyles among those who are married.

Married individuals were less likely to watch TV > 3 h/day than single and divorced ones in most age groups. This could be because married individuals have either increased hours of work or increased hours of house chores [37]. Married and divorced younger men and women were less likely to meet the PA guidelines levels. However, married women over 40 were more likely to meet PA levels.

The covariates in this study were included in the model to eliminate (adjust) the influence on the effect of the independent variable (marriage) on the outcomes. In other words, the observed effect of marriage (in the subgroups of age and sex) on physical activity and TV time is independent of the effects of the covariates on the outcomes.

Meeting PA minimum levels is the easiest way to prevent most non-communicable diseases and, although the information on its importance has piled up since the second half of the twentieth century, PA is frequently undervalued. In addition, the current global lifestyle imposes a higher SB to our daily life and provides a myriad of SB leisure-time activities such as watching TV. An approach to counteract the trends of PA and SB worldwide is to: (a) encourage systematically in workplaces, schools, and public places to break bouts of SB every couple of hours; (b) stimulate the augmentation of PA through simple lifestyle changes (e.g., using stairs instead of the lift, promote a standing-friendly culture at work and schools, stand on the public transport); and (c) providing public policies that encourage individuals to increase PA.

It is necessary to recognize our study limitations. Our sample represents the population from Brazilian capital cities and the federal district (urban areas); people living in these areas likely have lower levels of PA compared to rural areas not assessed [38]. Also, our study investigated leisure-time PA; people living in low-to-middle income countries have high PA in their occupation, for transportation (e.g., walking, cycling), and household chores [4], and the daily physical demands limits the individuals energy to perform leisure-time PA [39]. In addition, our data is from a self-reported telephone survey. Inaccurate reporting bias the results for reasons such as embarrassment, lack of interest, or the inability to report PA levels accurately. Also, it is possible that the quality of the relationships among married people could interfere in the results, and different experiences exist among unmarried people since they are not a homogenous group [5]. The Vigitel survey data is blinded, which precludes us from identifying duplicates and might result in inaccurate data. Finally, the absence of education level of participants and the socioeconomic condition in the model might have an influence in our results, as a higher educational level and socioeconomic condition are associated with higher leisure-time PA [40,41].

## 5. Conclusions

Our study found that viewing TV more than three hours a day was prevalent in more than a fourth of our sample, and there were no differences by sex or age. The highest prevalence was found in younger men and in older women. Married men and women were less likely to watch TV more than three hours a day in most age groups. Only 35% of participants met the PA recommended levels (>150 min/week), and younger married men and women are less likely to meet these levels. However, married women 40 years old or older were more likely to reach PA levels than single women. Our results suggest that the investments in public policies to encourage the practice of PA and reduction of TV time could be based on the marital status, sex, and age, prioritizing fewer active groups. Middle-aged single and divorced women should be prioritized to increase PA and reduce TV time.

## Figures and Tables

**Table 1 ijerph-19-00502-t001:** Prevalence of TV time, PA, marital status, age, and overweight of the sample by sex.

	All (*n* = 561,837)	Men (*n* = 212,440)	Women (*n* = 349,397)
TV > 3 h/day	25.2 (25.0; 25.4)	25.1 (24.7; 25.5)	25.2 (24.9; 25.6)
PA > 150 min/week	35.0 (34.7; 35.3)	43.0 (42.5; 43.4) *	28.2 (27.9; 28.5)
Age			
18–30	31.5 (31.3; 31.8)	35.1 (34.7; 35.6) *	28.5 (28.1; 28.8)
31–40	21.4 (21.1; 21.6)	21.2 (20.8; 21.6)	21.5 (21.2; 21.8)
41–50	17.8 (17.6; 18.0)	17.3 (17; 17.6)	18.3 (18; 18.5) *
51–60	14.5 (14.3; 14.6)	13.6 (13.3; 13.9)	15.2 (15; 15.5) *
61–70	8.6 (8.5; 8.7)	7.8 (7.6; 8.0)	9.4 (9.2; 9.5) *
71–80	4.5 (4.5; 4.6)	3.9 (3.7; 4.0)	5.1 (5; 5.2) *
80+	1.6 (1.5; 1.7)	1.2 (1.1; 1.3)	2.0 (1.9; 2.1) *
Marital Status			
Single	39.8 (39.6; 40.1)	42.1 (41.6; 42.5) *	37.9 (37.6; 38.3)
Married	38.5 (38.2; 38.8)	40.5 (40.1; 41.0) *	36.7 (36.4; 37.1)
Divorced	10.8 (10.6; 11)	11.7 (11.4; 12.0) *	10.0 (9.8; 10.3)
Widower	5.2 (5.1; 5.2)	1.5 (1.4; 1.6)	8.3 (8.1; 8.4) *
No reply	5.7 (5.6; 5.9)	4.2 (4.0; 4.4)	7 (6.9; 7.2) *

Data in prevalence (95% CI) calculated with a complex samples model to estimate the outcome for the population in 27 Brazilian Capitals from 2009 to 2019. * *p* < 0.05—Higher, sex differences.

**Table 2 ijerph-19-00502-t002:** Prevalence of TV time and PA by sex, age groups, and marital status.

	TV > 3 h/Day		PA > 150 min/Week	
	Men	Women	Men	Women
Age				
18–30	23.8 (23.0; 24.5)	24.7 (24.0; 25.3)	58.6 (57.8; 59.4) *	32.3 (31.6; 33.0)
31–40	26.2 (25.3; 27.1) *	21.5 (20.9; 22.2)	42.7 (41.7; 43.8) *	29.1 (28.4; 29.8)
41–50	24.8 (23.9; 25.7) *	21.9 (21.2; 22.5)	32.4 (31.5; 33.3) *	27.3 (26.6; 27.9)
51–60	25.7 (24.8; 26.7)	26.8 (26.1; 27.5)	29.9 (28.9; 30.8) *	27.7 (27.0; 28.4)
61–70	26.2 (25.1; 27.4)	32.1 (31.2; 32.9) *	30.3 (29.3; 31.5) *	24.9 (24.2; 25.6)
71–80	28.2 (26.7; 29.8)	34.5 (33.5; 35.5) *	27.7 (26.3; 29.1) *	19.0 (18.2; 19.8)
80+	27.9 (25.3; 30.6)	37.1 (35.4; 38.7) *	22.0 (19.8; 24.4) *	12.6 (11.6; 13.7)
Marital Status				
Single	24.9 (24.2; 25.5)	25.2 (24.7; 25.7)	54.4 (53.7; 55.1) *	30.9 (30.4; 31.5)
Married	23.8 (23.3; 24.4) *	22.2 (21.7; 22.7)	34.3 (33.7; 34.9) *	28.1 (27.7; 28.6)
Divorced	29.1 (27.8; 30.4)	27.8 (26.8; 28.8)	35.1 (33.8; 36.4) *	24.7 (23.7; 25.6)
Widower	31.0 (28.2; 33.9)	34.8 (33.9; 35.7)	26.0 (23.6; 28.6) *	20.4 (19.7; 21.2)
No reply	27.3 (25.4; 29.4)	26.5 (25.5; 27.6)	40.1 (38.0; 42.3) *	27.9 (27; 28.9)

Data in prevalence (95% CI) calculate with a complex samples model to estimates outcomes for the population in 27 Brazilian Capitals from 2009 to 2019. * Higher significantly, sex differences.

**Table 3 ijerph-19-00502-t003:** Odds Ratio (OR 95% CI) of TV time and PA levels for married by age groups and sex.

Age	Reference Group	TV > 3 h/d		PA > 150 min/Week	
		Men	Women	Men	Women
18–30	Single	1.00 (0.88; 1.14)	0.99 (0.89; 1.10)	0.59 (0.53; 0.66) *	0.74 (0.67; 0.81) *
	Divorced	0.65 (0.53; 0.78) *	0.69 (0.60; 0.79) *	0.96 (0.81; 1.14)	1.26 (1.09; 1.45) *
31–40	Single	0.79 (0.71; 0.88) *	0.75 (0.69; 0.82) *	0.79 (0.72; 0.87) *	0.93 (0.86; 1.00)
	Divorced	0.73 (0.63; 0.84) *	0.72 (0.64; 0.82) *	1.03 (0.91; 1.17)	1.11 (0.99; 1.24)
41–50	Single	0.66 (0.58; 0.75) *	0.68 (0.62; 0.75) *	0.98 (0.88; 1.09)	1.29 (1.18; 1.40) *
	Divorced	0.93 (0.80; 1.07)	0.71 (0.63; 0.81) *	1.14 (1.01; 1.28) *	1.21 (1.08; 1.35) *
51–60	Single	0.80 (0.69; 0.93) *	0.66 (0.60; 0.73) *	0.93 (0.80; 1.07)	1.36 (1.24; 1.49) *
	Divorced	0.88 (0.75; 1.03)	0.81 (0.71; 0.93) *	1.24 (1.07; 1.42) *	1.13 (0.99; 1.30)
61–70	Single	0.73 (0.58; 0.92) *	0.65 (0.58; 0.73) *	0.98 (0.81; 1.19)	1.44 (1.29; 1.60) *
	Divorced	0.76 (0.61; 0.93) *	0.79 (0.66; 0.95) *	1.24 (1.03; 1.50) *	1.21 (1.00; 1.47) *
71–80	Single	1.00 (0.72; 1.38)	0.84 (0.73; 0.98) *	1.28 (0.90; 1.82)	1.36 (1.15; 1.61) *
	Divorced	0.87 (0.62; 1.20)	0.79 (0.66; 0.95) *	1.30 (0.99; 1.71)	1.44 (1.01; 2.04) *
80+	Single	0.83 (0.41; 1.67)	0.59 (0.45; 0.79) *	1.16 (0.59; 2.27)	0.91 (0.61; 1.36)
	Divorced	1.00 (0.60; 1.66)	0.92 (0.67; 1.26)	1.05 (0.61; 1.81)	1.09 (0.46; 2.59)

Complex Sample Logistical Regression adjusted by year of data collection, obesity, hypertension, diabetes, health status and smoking as covariates. CI: confidence Interval. OR: (CI 95%) * *p* < 0.05.

## Data Availability

The data that support this article can be found at the Brazilian Ministry of Health website at http://svs.aids.gov.br/download/Vigitel/ (accessed on 2 January 2022) from the Surveillance of Risk and Protection Factors for Chronic Diseases by Telephone Survey—VIGITEL section.

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
