# Peer review of "Age and Sex-Related Associations between Marital Status, Physical Activity and TV Time"

_ijerph, 2022, doi:10.3390/ijerph19010502_

Round 1

Reviewer 1 Report

Overall considerations

The manuscript titled “Age and Sex-Related Associations Between Marital Status, Physical Activity and TV Time” deals with an interesting topic about sedentary behavior. The authors investigated the correlation between age, obesity, sex, marital status, physical activity levels, and TV time. The results highlight the main differences between the categorical classes. This paper is interesting and suitable with the remit and purpose of the journal, even if minor concerns need to be solved before the publication.

Abstract

In the abstract section, authors could add a sentence about the investigation problem rather than reporting the aim already. Secondly, they could avoid the repetition of “We”. Third, the authors may add a conclusion sentence.

Keywords

Marriage, gender, and age groups could be avoided since they do not discriminate the paper's aim. They could add “physical activity”, “health prevention”.

Introduction

L44-45 Authors could clarify this sentence because it is not clear.

L63 Try to rewrite this sentence and use numbers instead of words because “Twenty seven and a half percent” is hard to understand.

L80-82 Authors could add more emphasis to the study’s aim, such as answering to “why are we conducting this study? What are the main outcomes we expect?” because written in this way, it takes a back seat.

Results

I would suggest elaborating a figure to make results more communicative.

Discussion

L208-209 Reference is needed to state this.

Authors could add a section highlighting the importance of physical activity and how the SB-trend could stop by suggesting practical approaches.

Conclusion

Try to highlight better the scientific/clinical relevance of the work. Please provide a clear message of the importance of this paper in the scientific community.

Tables

Table 1 and 2 Authors have to clarify to which statistical approach p-value refers to.

Minor comments

Several typos and double spaces are present. Revisit the final version.

Author Response

Kind regards

Reviewer 2 Report

After reviewing the manuscript entitled “Age and sex-related associations between marital status, physical activity and TV time” my opinion is that it requires major revisions.

Comments

In general, this is an interesting paper. The authors evaluate age and sex-related associations between marital status with PA and TV time. However, while reading the manuscript I came across several issues that need to be addressed before I believe the manuscript is suitable for publication. Please see below my suggestions.

Abstract

- Total age range and age groups must be added.

Introduction

- The associations between marital status and health benefits reported by the authors are repetitive (lines 46 & 64-65).

- The authors must rewrite some parts of this section (e.g., lines 73-74; 78-79). Moreover, there are several word spacing errors (e.g., lines 48, 54 & 64).

Materials and methods

- The authors must improve the sample selection methods. Complete information about CI (%), sampling error (%), clusters, quotes and/or stratification used is needed.

- Lines 92-95. The sentence “The system establishes a minimum sample size of approximately 2000 individuals in each city to estimate the frequency of the main risk factors for chronic non-communicable diseases (hypertension, diabetes, and dyslipidemia)” must be clarified. What system? Why is it necessary a minimum of 2000 subjects per city? A justification is required.

- Line 112. Include sex and year of data collection as covariates (they are mentioned in the statistical analysis section; lines 129-130).

Results

- Could the authors provide more information about the logistic regression model used to estimate the ORs? How does each covariate influence ORs reported in Table 3? The authors must explain at least the influence of hypertension and obesity (the most prevalent covariates) on ORs.

Discussion

- Some content of this section seems to be speculative (e.g., lines 221-222).

- Lines 226 - 227. Why do authors consider “self-efficacy” as an external factor?

- Line 227. Write “findings” instead of “finds”.

- Line 229. Delete “and”.

- Study limitations. The authors should discuss about socio-economic status and education level of participants since these are key factors influencing PA and also sedentary behaviours.

Conclusions

- Line 259. Write “married women” instead of “women”.

All these comments must be carefully addressed and included in a new version of the manuscript.  

Author Response

Kind regards.

Reviewer 3 Report

Thank you for the opportunity to review your paper.
It's great that you have large amounts of data.
However, I have some questions.

Major points
(1) This paper tried to clarify the prevalence and associations of PA and TV time. PA and TV time were assessed by interviews. Therefore, your main outcomes were inaccurate information because they included bias and error. How valuable were the findings of this study, especially regarding the prevalence and some associations? If you use VIGITEL data, you should conduct a validation study between the interviews and accelerometers for PA. A reliability study concerning PA and TV time is also needed. I recommend pre-verification of assessment methods for this epidemiologic study. In addition, how was the interviewer trained? Please add this information.

(2) Brazil is no exception, Lifestyles (PA and TV watching) are changing with the advancement of technologies such as the Internet. This study used long-term data (from 2009). How should we interpret these findings involving old data? How can they be generalized to the current and future use of technology? Additionally, please suggest how to utilize these findings in policy implementation. 

Minor points
[line 84] Could you add a figure about participants’ flow in this study?

[line 89] How did you consider duplicate information from those who participated in multiple years? The data were also surely affected by aging.

[line 99] Could you add an explanation of and the citation for the definition of PA and TV time? Why did you use >150 min/week as the PA cut-off? >=150 min/week? >=150 and <=300 min/ week? I also want to know how you selected the cut-offs for TV time.

[line 129] Do you have data about the residences in the area(city)? 

[Table 1] Why did you categorize 80 years of age? 81+? Please recalculate all statistical information (Tables).
In the footnote, why did you show data from 2006 to 2016? Did you explain this in the methods?

[Table 3] Why did you not show the prevalence of PA and TV time in Table 3?
Please add information about the analysis model, including covariates, in a footnote.

[Tables] There were 'male', 'female', 'men', and 'women' at the Tables. Please check the consistency of all terms in the paper.

Author Response

Kind regards.

Round 2

Reviewer 2 Report

Once the authors have amended the manuscript in accordance with my comments I think that it can be published.

Author Response

Best regards.

Reviewer 3 Report

I appreciate your efforts.

Comment 3
I understand the reason for the missing of the study flow. 
Could you show the breakdown samples (as a supplemental table) for each survey (year)? 
I think the readers want to know the sampling bias among all surveys in your study.

Comment 4
Could you explain the concerns of the duplicate information in the limitation section?

Comment 8
I saw the odds of PA and TV among small subgroups based on three factors (marital status, age, and sex) in Table 3. Table 2 were showed by age or marital status. Are Table 2 and Table 3 the same prevalence?
In addition, you explained the reference group: single in logistic regression in 2.4. Statistical analysis. However, I cannot find all marital statuses in Table 3. How did you use single, married, divorced, widowed, and no reply in the study? I cannot find the marital status section in the methods.

Author Response

Best regards.
